# A Practical Evaluation on RSA and ECC-Based Cipher Suites for IoT High-Security Energy-Efficient Fog and Mist Computing Devices

**DOI:** 10.3390/s18113868

**Published:** 2018-11-10

**Authors:** Manuel Suárez-Albela, Paula Fraga-Lamas, Tiago M. Fernández-Caramés

**Affiliations:** Department of Computer Engineering, Faculty of Computer Science, Universidade da Coruña, 15071 A Coruña, Spain; m.albela@udc.es

**Keywords:** ECC, ECDSA, RSA, IoT, IoT security, energy efficiency, mist computing, edge computing

## Abstract

The latest Internet of Things (IoT) edge-centric architectures allow for unburdening higher layers from part of their computational and data processing requirements. In the specific case of fog computing systems, they reduce greatly the requirements of cloud-centric systems by processing in fog gateways part of the data generated by end devices, thus providing services that were previously offered by a remote cloud. Thanks to recent advances in System-on-Chip (SoC) energy efficiency, it is currently possible to create IoT end devices with enough computational power to process the data generated by their sensors and actuators while providing complex services, which in recent years derived into the development of the mist computing paradigm. To allow mist computing nodes to provide the previously mentioned benefits and guarantee the same level of security as in other architectures, end-to-end standard security mechanisms need to be implemented. In this paper, a high-security energy-efficient fog and mist computing architecture and a testbed are presented and evaluated. The testbed makes use of Transport Layer Security (TLS) 1.2 Elliptic Curve Cryptography (ECC) and Rivest-Shamir-Adleman (RSA) cipher suites (that comply with the yet to come TLS 1.3 standard requirements), which are evaluated and compared in terms of energy consumption and data throughput for a fog gateway and two mist end devices. The obtained results allow a conclusion that ECC outperforms RSA in both energy consumption and data throughput for all the tested security levels. Moreover, the importance of selecting a proper ECC curve is demonstrated, showing that, for the tested devices, some curves present worse energy consumption and data throughput than other curves that provide a higher security level. As a result, this article not only presents a novel mist computing testbed, but also provides guidelines for future researchers to find out efficient and secure implementations for advanced IoT devices.

## 1. Introduction

The rise of the Internet of Things (IoT) paradigm is expected to connect to the Internet more than 30 billion devices by 2020 [1]. This unprecedented number of devices, along with their estimated fast growth, raises two main concerns: security and energy consumption. The access and communication flexibility that IoT brings come with numerous threats that introduce the need for computational and energy-demanding security mechanisms. Since IoT allows sensor and actuator devices to communicate end-to-end through the Internet, the performance and security techniques implemented by IoT end-nodes impact the rest of the involved devices. From fog gateways to cloud and backend infrastructures, along with routers and network devices, they are all affected by the way IoT end-nodes behave. Regarding energy consumption, it is estimated that the United States data centers will consume approximately 73 TWh in 2020 [2]. A similar report from the European Union [3] estimates that data center energy consumption in Europe reached a peak of 102 TWh in 2017 and it will slowly decay to 87 TWh in 2020. New disruptive developments such as cryptocurrencies can also have a massive impact on the expected trends of energy consumption. For instance, in 2014 the energy consumption of the whole Bitcoin infrastructure was estimated to be equal to the electricity consumption of Ireland [4].

Nowadays, IoT devices are present in numerous environments, such as in agriculture [5], home automation [6,7], transportation [8], telemetry [9,10], Industry 4.0 [11,12], defense and public safety [13], blockchain-based applications [14,15], energy-efficiency applications [16,17], augmented reality [18,19] or in healthcare [20], where security issues can pose risks for human safety and privacy [21]. Nevertheless, IoT security is still a challenge and a major concern [22], with several examples in the literature of insecure cyber-physical systems [23,24,25,26], resulting in a slowdown in the IoT adoption due to the perception of insecurity [27]. Most of the authors that analyze these insecure systems conclude that the main problem is addressing security as a secondary requirement instead of building the systems with security in mind since their inception [23,24,25,26].

To guarantee a secure and energy-efficient deployment of IoT networks, three main elements must be considered and addressed as a whole: IoT architecture, IoT hardware and the required security mechanisms of all the involved devices. Due to the advances in System-on-Chip (SoC) efficiency, current IoT nodes have enough computational capabilities for processing most of the data and unburden the devices of higher layers from data processing. This movement of data processing capabilities from the gateways to the end devices is called mist computing [28]. More powerful and efficient IoT end-nodes allow not only for reducing the data needed to be transmitted to higher layers, but also to implement more complex services and strong end-to-end security mechanisms. While under the fog computing paradigm the amount of data per transaction grows as we go to upper layers [29,30,31], in a mist computing architecture data are kept as close as possible to where they are originated, providing the ability of serving complex services directly to end devices. Therefore, a hardware and software implementation that considers the different devices involved on fog and mist architectures will ensure greater security and efficiency levels than previous approaches that treated separately each layer of the architecture.

This article includes three main contributions aimed at creating a testbed for determining the impact of securing fog and mist communications. First, in order to establish the basics, it presents a high-security fog and mist computing architecture and an analysis of the main and the latest hardware platforms to be used as mist end devices. Second, it thoroughly explains the design and implementation of a testbed that allows for obtaining energy and throughput measurements when securing the communications of fog and mist devices. To maintain the interoperability between all the layers, Hypertext Transport Protocol Secure (HTTPS) was selected as the communication protocol. The latest Elliptic Curve Cryptography (ECC) and Rivest-Shamir-Adleman (RSA)-based Transport Layer Security (TLS) cipher suites are analyzed and only the ones aligned with the requirements of the next TLS standard (i.e., TLS 1.3) are selected. Third, multiple tests are conducted to determine the performance and energy-consumption impact of using TLS with different RSA and ECC implementations in a real-world scenario. It is performed an analysis in terms of security level, scalability, energy consumption and data throughput in two different scenarios: a fog gateway to mist end-device communication and a direct mist end-device to mist end-device communication.

The rest of this paper is structured as follows. Section 2 presents the state-of-the-art hardware platforms and describes the main characteristics of fog and mist computing systems, and their main security challenges. Section 3 details the selected hardware, the implemented testbed and its architecture. In Section 4, the performance of the testbed is evaluated in terms of energy consumption and data throughput. Finally, Section 5 is devoted to conclusions.

## 2. Related Work

### 2.1. From Fog to Mist Computing

Cloud-centric architectures allow for providing complex services with high demands on computational power, data storage and network bandwidth that are difficult to accommodate on personal computers or Small and Medium-sized Enterprise (SMEs) mainframes. Although the emergence of cloud computing solutions democratized the access to High Performance Computing (HPC) capabilities, its centralized nature comes with restrictions that are difficult to accommodate on real-time IoT systems. To alleviate the inherent high latency of cloud computing, edge-centric approaches [32,33] move the computational resources closer to the data generation and service access. Fog computing has brought data processing and service provision to intermediate network layers (i.e., fog gateways), using the highest layers of the architecture strictly when needed [29]. This allows for a reduction on both latencies and bandwidth requirements, decreasing the deployment cost, improving energy efficiency and accelerating service responses. It is important to note that, in the past, the use of gateways was needed because IoT end devices did not have enough computational power to process the data collected from sensors and actuators or to implement the security mechanisms required for providing direct access. Thus, IoT fog systems usually isolate end devices by providing access to them through gateways, which are in charge of translating end-device communications protocols and implementing security mechanisms. Such a dependence on gateways for processing end-device data and for providing security to data transmission, emptied the bottom layer of the architecture from high-security and real-time actuation capabilities. Mist computing moves the computational resources to the end devices, thus providing them with the capability of processing sensor and actuator data and requiring implementation of high-security mechanisms and standard communications technologies. In addition, mist end devices are powerful enough to access other end-device information and directly provide lightweight services. The result is end-to-end communications and very low latencies for IoT deployments.

An example of generic IoT edge architecture with fog and mist computing layers is shown in Figure 1. With the computational capabilities of mist end devices, both data processing and service provisioning can be performed on the lower layer of the architecture, alleviating higher layers from such tasks, as pointed out by the arrow in the left of the figure. Only the most complex processes or the ones that need to integrate data for a large set of sources are delegated to fog gateways or to the cloud when needed. Another important difference with a typical fog architecture is the ability of the mist end devices to communicate directly with each other, thus reducing the network traffic of the top layers.

### 2.2. Hardware and Software for Mist Computing Devices

In a mist computing deployment, the selected end-device hardware platform is essential. To allow for implementing standard Internet communications protocols such as Hypertext Transport Protocol (HTTP) or Message Queuing Telemetry Transport (MQTT) [34], IoT hardware platforms must be equipped with relatively high bandwidth communications interfaces. In [35] it is provided a comprehensive comparison among different wireless communications technologies for IoT, being Wi-Fi the one that presents the best results in terms of data-rate, energy efficiency, and security for distances between 1 to 50 m. The ability to provide real-time services directly from the end-device layer is also a requirement for mist computing. Due to this fact, clock speed and RAM requirements, while constrained to improve energy efficiency, are higher for mist than for fog IoT sensor networks. Taking the mentioned limitations into account, several hardware platforms emerge as possible candidates to be used in mist deployments (the main characteristics of some of the most popular are summarized in Table 1).

Due to the resource-constrained nature of these platforms, it is usually not possible to execute the necessary cryptographic algorithms on the main SoC processor for securing mist devices. In addition, hardware acceleration is often required for maintaining acceptable energy consumption and throughput values when executing algorithms involved in public-key cryptography such as Elliptic Curve Digital Signature Algorithm (ECDSA) [42]. Speed-ups can also be achieved by using specific hardware for the execution of less demanding cryptographic algorithms such as block ciphers [43] or hash algorithms [44], which are also required for securing IoT network communications.

It is important to note that even if a platform embeds a hardware-acceleration module into the SoC for performing certain cryptographic functions, software implementations that make use of the available hardware acceleration are also needed. Software optimizations to tackle specific hardware accelerators can impact significantly both the algorithm execution time and the energy consumption [45].

The ESP32 is the only hardware board in Table 1 that fits all the previously mentioned requirements. The ESP32 embeds a dedicated crypto-engine for accelerating the execution of different cryptographic algorithms. The other hardware platform from the ones compared in Table 1 that presents hardware-acceleration capabilities is the Arduino MKR WiFi 1010, but only SHA-256 acceleration and ECC508 Crypto Authentication is provided. In comparison, the official ESP32 Software Development Kit (SDK) implements the required software functions to make hardware acceleration available for high-level protocols such as TLS, and supports RSA, Secure Hash Algorithm (SHA), Advanced Encryption Standard (AES) and ECDSA. These features make the ESP32 the best candidate among the platforms compared to implement a mist end-device. The main hardware and software characteristics of this board are further discussed in Section 3.1.

### 2.3. Main Security Threats for Mist Computing Devices

The main information security characteristics are confidentiality, integrity and availability, also known as the CIA triad. Each one of them grants different properties to a communications system:
Confidentiality: it disallows unwanted access to sensitive information.Integrity: it protects the transmitted data for unwanted modification or tampering.Availability: it is related to the ability of accessing the data when they are needed.

Different types of attacks can be performed to break one or several of these security characteristics when targeting mist deployments. Depending on which part of the communications architecture is targeted by the attack, they can be classified as [46]:
Physical attacks: they target the physical device or the physical communication channel:
–Node tampering: it consists on altering the mist device itself or the sensors or actuators connected to it to access or modify sensitive information or the device behavior [47].–Jamming: an attacker blocks mist device wireless communications by transmitting with a jammer device [48].–Radio Frequency (RF) interference: it consists in introducing devices on the network that generate signals or noise that interfere legitimate communications [49].–Malicious node injection: it consists on adding a malicious mist node to the mist layer with the intention of introducing tampered data or to access the data being transmitted between the victim mist devices [50].–Physical damage: an attacker can prevent a mist node from working properly or can neutralize it completely by totally or partially destroying it.–Malicious code injection: by physically accessing the mist node an attacker could change the code controlling the node and achieve total access to the mist layer [51].–Sleep deprivation attack: an attacker tries to maximize the energy consumption of a node, which can have a massive effect on the overall energy consumption of the mist layer and thus reduce the life of battery-operated nodes [52].Network attacks: they try to exploit some vulnerability on the way the communications are established on the mist layer:
–Traffic analysis attacks: the attacker can somehow intercept the traffic transmitted over the mist network and analyze it to infer any type of useful information about how the network and the mist devices work [53].–Man-in-the-Middle (MitM) attacks: one of the most common network attacks where a third party captures the communications between two legitimate mist nodes and accesses or tampers the transmitted data [54].–Denial of Service (DoS) attack: an attacker tries to prevent legitimate access to mist nodes by flooding the mist layer communication network with fake traffic or by tampering with the normal functioning of the network [50].–Message replaying attack: an attacker can capture a command sent to a mist device and replay it at its convenience [55].–Sybil attack: a malicious mist node can spoof the identity of other legitimate nodes and act as them [56].Software attacks: malware attacks using viruses, worms or malicious scripts targeting mist devices:
–Virus and Worms: attacks that consist on some type of code that can take partial or total control of the mist end-device firmware, and that can be easily replicated to other mist nodes in an autonomous way [57].–Malicious scripts: mist nodes are usually capable of executing arbitrary pieces of code. By introducing a malicious script, an attacker can gain control of a mist node while the node seems unaltered to the rest of the mist network.–DoS attacks: a DoS attack can be also performed at a software level, by disallowing the access to a service at an application level. It is usually performed by gaining control of mist nodes or fog gateways that provide such a service.Encryption attacks: the last type of attacks that can be performed against mist devices are the ones that try to break the encryption of the communications (in case encryption techniques are used to secure communications) by obtaining the private key of one of the involved parties:
–Side-channel attacks: the attacker uses behavioral information of the device that is applying the cryptographic algorithms. By measuring the elapsed time required for signing or verifying on a public-key cryptographic scheme, or the energy consumption of a mist device during certain phases of the cryptographic protocol, an attacker can infer information and guess the encryption key being used without having a direct knowledge of the plaintext or the ciphertext [58].–Cryptanalysis attacks: an attacker can obtain the encryption key by obtaining the plaintext or the ciphertext [59].–MitM attacks: the attacker can intercept the communications between two legitimate parties when the key negotiation phase is performed. At such a point, the attacker would be able either to establish a spoofed shared secret or to obtain the negotiated shared secret and decrypt or tamper any messages transmitted between the two parties [60].

### 2.4. Impact on Energy Consumption of Securing Mist End-Device and Fog Gateway Communications

As of writing, no references were found in the literature about the analysis on the impact of security on a mist computing environment. As it was indicated previously in Section 2.1, the ESP32 hardware platform can fulfill the requirements of a mist deployment and, although it has not been found specific literature on such an application of the platform, it has been already evaluated by different authors in other fields. For instance, in [61] the characteristics of the ESP32 are described and compared with other platforms. Such a paper describes the most relevant characteristics, but no details are given on the available cryptographic hardware acceleration. The paper also presents a practical application of the ESP32 module by implementing an oscilloscope, and concludes that the module is an excellent option for creating IoT devices. Similarly, other authors describe the implementation of a web server on an ESP32 for real-time monitoring of a photovoltaic system [62]. In such a paper the researchers demonstrate that the implemented system allows for obtaining real-time data on the voltage and the current from the monitored system, but they do not take advantage of the ESP32 cryptographic acceleration or implement any security on the communications. Another implementation of an IoT system based on the ESP32 is an emergency response system for fire hazards [63]. The only reference to security the authors indicate is related to the use of MQTT, but it is referred to the data security mechanisms provided by the protocol itself, which actually needs to run over TLS to provide privacy and data integrity [34].

Regarding energy efficiency, some authors conclude that moving the processing power closer to the data origin could yield energy efficiency improvements [64], but they remark that there is a great dependency with system design factors, such as the ratio between active time and idle time, the number of downloads from other users or the number of uploads. This fact indicates that empirical tests should be performed to determine the actual gains on a real system.

There are other works focused on improving energy efficiency of IoT systems [65,66] or on improving energy efficiency of other systems using IoT solutions [67], but they do not address the impact of security mechanisms on energy consumption, although security requirements should be the core of any IoT system, and especially of the mist layer.

### 2.5. Security Levels

When the strength of a cryptographic algorithm is assessed, the main parameter that is usually considered is its key size in bits. Using the same algorithm, larger key lengths make brute force attacks more difficult, thus creating more secure systems. One important consideration is that key size defines an upper-bound on the security an algorithm can provide, but the actual strength of the algorithm can be lower: the security level that a certain algorithm provides is a way of measuring the effort needed to break its trapdoor function [29]. Generally speaking, symmetric-key algorithm security level has a direct correspondence with the selected key size (i.e., the greater the key size, the more secure the system is), but asymmetric-key algorithms security level is lower than the key size employed.

The relationship between key size and security level of different asymmetric-key algorithms varies greatly, as it can be observed in Table 2, where the security levels for different key sizes of RSA and ECC curves are compared. Moreover, the relationship between security level and key size of a certain algorithm does not have to grow linearly, like the case of RSA. Therefore, it can be concluded that relying on the key size for comparing different algorithms, or even the same algorithm, like the case of RSA, may be misleading. For a fair comparison between two algorithms in terms of energy efficiency and throughput, key sizes that provide roughly the same security level in both cases must be selected. For this reason in this paper the concept of security level is used for comparing the proposed algorithms and to present the results for the experiments detailed in Section 4.

### 2.6. ECC Curve Implementations for Resource-Constrained Devices

The ECC curves used by modern cryptosystems are implemented over two number fields. The prime field $F_{p}$, where *p* is a prime number, and the binary field $F_{2^{m}}$, where the points of the curves are represented as a polynomial that can be expressed as a binary number (e.g., the polynomial $x^{4}+x^{2}+1$ is represented by the binary number {1 0 1 0 1}). For both fields, *p* and $2^{m}$ indicate, respectively, the number of the elements in the field. Binary field curves present worse performance and energy efficiency when executed on general-purpose processors [68], although they can outperform prime field curves when using dedicated hardware [69]. They can also run faster and more efficiently on general-purpose processors with optimized instruction sets for accelerating some calculations for binary field curves [70]. Some concerns about possible security issues of binary field curves [71] make them not adequate for certain uses. Moreover, TLS libraries for resource-constrained devices (e.g., mbed TLS [72]) lack the necessary implementations for using this type of curves [73].

Regarding prime field curves, there are two main subgroups that differ on how the domain parameters of the curve are generated. There are curves generated with verifiable random domain parameters (e.g., secp192r1) and curves that are generated with domain parameters over a prime field associated with a Koblitz curve (e.g., secp192k1). A detailed explanation on how the domain parameters are obtained for each curve and its implications for both prime and binary fields can be found in [74].

The most extended curves available for TLS communications are the ones defined by the National Institute of Standards and Technology (NIST) [75]. The problem with these curves is the algorithm used to generate the pseudo-random numbers needed to define their domain parameters (i.e., the Dual_EC_DRBG (Dual Elliptic Curve Deterministic Random Bit Generator)), since there were suspicions of a possible backdoor [76].

Another largely available set of curves is the one defined by the Standards for Efficient Cryptography Group (SECG) [77], which uses domain parameters over a prime field associated with a Koblitz curve. The main theoretical advantage of these curves is that their generation process does not rely on a pseudo-random number generator, thus avoiding the possible backdoors that the NIST curves are suspicious of having. Moreover, Koblitz curves have some properties that allow for accelerating certain ECC calculations over them.

Finally, another option is to use the so-called Brainpool curves [78]. This Internet Engineering Task Force (IETF) [79] standard is aimed at providing a set of curves with parameters generated in a pseudo-random, yet completely systematic and reproducible way, which can resist current crypto-analytic attacks. The main limitation of these curves is their performance [80]. Due to the way the random primes used in Brainpool curves are generated, there is no fast reduction possible to these curves, in contrast to NIST and SECG curves. Therefore, although these curves are supposed to avoid possible security issues of other curves, their limited performance makes them not appropriate for resource-constrained devices.

## 3. Design and Implementation of the Fog and Mist Computing Testbed

The general architecture of the designed testbed is shown in Figure 2. It consists of two different layers: a mist layer with two mist end devices and a fog layer with a fog gateway. To enable the communications among the different devices, a communications gateway is placed between the fog and the mist layers. The communications gateway can be any device capable of allowing a relatively large number of mist devices to access the fog gateway and the upper layers of the architecture (e.g., a Wi-Fi router, femtocell or picocell base stations). In the presented testbed, a Wi-Fi router is used by connecting it to the fog gateway through an Ethernet cable and to the mist end devices through Wi-Fi. A monitoring subsystem is used to execute the experiments and obtain energy consumption measurements. Such a monitoring subsystem is formed by a Single Board Computer (SBC) and current sensors that allow for obtaining the current consumed by each of the devices of the fog and mist layers during the tests performed in Section 4. In the case of the mist end devices, no transducers were embedded since the transducer energy consumption and communications capabilities could bias the obtained results. Two different scenarios were set to illustrate two types of communications: fog gateway to mist end-device communications and mist end-device to mist end-device communications.

It is important to note that, although the testbed architecture was designed to measure the energy consumption and throughput of devices of the fog and mist layers, it can be easily extended to cover other scenarios. For instance, if it is required to measure a higher number of devices it would only be necessary to deploy more SBCs with their corresponding current sensors. Moreover, in case of needing to perform the same measurements when the network presents long or unpredictable latencies between the SBCs (e.g., to measure the energy consumption of IoT nodes deployed on separated geographical locations or in scenarios where communications must go through several network layers), advanced time synchronization approaches can be implemented. In the case of not being able to use Network Time Protocol (NTP) servers, Global Positioning System (GPS) time synchronization (with the simple addition of a GPS module to the SBCs) could be used to provide a common time reference to all the SBCs. When time synchronized, each of the SBCs could start and stop the tests and the energy measurement procedure in the same way as the implemented testbed presented in this paper.

### 3.1. Selected Hardware

To perform the designed experiments, the devices in the fog layer and the mist layer (i.e., two end devices and a gateway) must be able to support the same cipher suites, ECC curves and RSA key sizes. For the sake of fairness, the selected devices for the hardware testbed must be constrained in terms of energy consumption, but, at the same time, be able to execute the required cryptographic algorithms. The traffic and data throughput requirements for a fog gateway usually imply the use of a device powerful enough to run a Linux distribution, thus supporting well known and extensively used TLS implementations such as OpenSSL [81]. As it was previously indicated in Section 2.2, the options for building a mist end-device are limited, since most IoT end-nodes do not support TLS or, if they do, they support only a few insecure cipher suites.

Due to the previously mentioned requirements, an ESP32 module was selected as mist end-device. It is an IoT platform that embeds an Xtensa dual-core 32-bit LX6 microprocessor (Tensilica, San Jose, CA, USA) that runs at 240 MHz and has 520 KB of Static RAM (SRAM). With a peak energy consumption of only 1.65 W, it presents a great balance between computational power and energy efficiency. It features both Wi-Fi and Bluetooth v4.2 BR/EDR and Bluetooth Low Energy (BLE) interfaces. The Wi-Fi interface supports IEEE 802.11 b/g/n modes of operation and WPA/WPA2 authentication. It provides several buses and communication interfaces that are ideal for IoT sensor nodes, such as I2C, UART and programmable GPIOs. Moreover, a crypto-engine is integrated into its SoC, thus providing hardware acceleration for different cryptographic algorithms such as ECC, RSA, AES and SHA, which are all used by the cipher suites tested in Section 4.

A SBC was chosen to implement the fog gateway, since it presents a good trade-off between energy efficiency, computational capabilities and software implementation flexibility. Specifically, Orange Pi PCs [82] (Shenzhen Xunlong Software CO., Limited, Shenzhen, China) were selected because, as explained in [29], they provide good performance at a reduced cost when compared to similar alternatives. Every Orange Pi PC embeds 1 GB of Double Data-Rate (DDR) RAM and an Allwinner H3 SoC, which includes an A7 quad-core 1.6 GHz processor. Regarding the SBC’s communication capabilities, an integrated 100 Mbit Ethernet interface allows for transmitting large amounts of data over the Internet or on a Local Area Network (LAN). The Orange Pi PC also includes a High-Definition Multimedia Interface (HDMI) video output and three Universal Serial Bus (USB) 2.0 connectors for connecting peripherals. In addition, the SBC provides 40 General-Purpose Input/Output (GPIO) pins that support communications buses such as Inter-Integrated Circuit (I2C) or Universal Asynchronous Receiver-Transmitter (UART), which can be used to communicate with sensors or other devices. The available driver for the Allwinner H3 SoC still does not support hardware acceleration for RSA or ECDSA [83], so it was not possible to use it during the tests presented in this article.

An Asus RT-N12 D1 Wi-Fi router was used as the central communication device. The Orange Pi PCs were connected using Ethernet Cat 5e patch cables, while the ESP32 was connected through the Wi-Fi interface. For powering up the ESP32 and the Orange Pi PC gateway, two dedicated 5 V/2 A power sources were used. Due to the ESP32 reduced energy consumption, it is possible to use batteries to power it up, but, in order to provide a stable and constant energy source, a dedicated power supply was used instead.

To obtain the energy consumption values external current sensors were used (Adafruit INA219 current sensors). A second Orange Pi PC was also used as the device in charge of obtaining the energy consumption values from the INA-219 current sensors during the tests. The INA219 can operate in different modes: with 0.1 mA steps for a maximum of ±400 mA or 0.8 mA steps and a maximum of ±3.2 A. The maximum voltage allowed by the sensor is 26 V, far beyond the 5 V of the power supplies used for powering up the mist device and the fog gateway. The different values measured by the sensor can be accessed through the I2C bus, which eases the communication with the Orange Pi PC in charge of obtaining the energy consumption measurements.

Figure 3 shows the main components of the testbed. Two different configurations of the testbed were used, whose involved components are inside two dotted rectangles. The setup at the top (A) represents a mist end-device to fog gateway communication scenario, while the one at the bottom (B) is for mist end-device to mist end-device communications. Only two 5 V power supplies are used. The laptop is the device in charge of starting and stopping the tests and of persisting the energy and throughput measurements for further analysis. As an example, Figure 4 shows the main components of the testbed for the scenario A (for the sake of clarity, the Ethernet connections depicted in Figure 3 are not shown in Figure 4).

### 3.2. Software

This Section details the main software used for implementing the testbed and executing the experiments. For programing the ESP32 modules, the ESP32-IDF [84] release/v3.1 SDK was used. The SDK was installed on a Windows PC under a MINGW environment [85] with OpenSSL 1.0.2n. The ESP32-IDF is the official development framework for the ESP32 and it is based on the real-time operating system FreeRTOS [86]. To enable the execution of the different tests, two FreeRTOS applications were developed for the ESP32. The application for the ESP32 that acts as mist server consisted on an HTTPS server that made available several JSON files with different payload sizes (described later in Section 4). The second application consisted on an HTTP and an HTTPS client. The HTTP client is used to start and stop the measuring procedure and the HTTPS client allows for downloading the JSON files from the ESP32 module that acts as a mist server. The underlying TLS implementation is carried out through the mbed TLS library (version 2.9.0) [72], a library aimed at making TLS fully available on embedded devices.

The Linux distribution ARMBian [87] was installed on both Orange Pi PCs. Python [88] was used to program the code for running the tests. Different Python scripts were used for remotely starting and stopping the energy measurement procedure by accessing the INA219 current sensors through the I2C bus. A Python wrapper was programed over the pi-ina219 library (version 1.2.0) [89]. The wrapper allowed for reading two INA219 sensors over the same I2C bus and for launching an HTTP server that can start and stop remotely the energy measurement procedure. The tested maximum sampling rate for two sensors was 1000 Hz, resulting on a 500 Hz effective sampling rate for each of the sensors during the tests.

Finally, a virtual machine with Debian was deployed on the laptop. The purpose of this machine is to allow for accessing the testbed during the execution of the tests through an SSH connection. The same virtual machine was used to download all the generated data to analyze them and to generate the different charts presented in Section 4.

### 3.3. Selected Cipher Suites and Certificate Generation

Two different cipher suites were selected for the tests: One based on RSA and another one based on ECDSA (for signing certificates). The remaining algorithms involved in the cipher suite (e.g., key exchange, block cipher, Hash Message Authentication Code (HMAC)) were identical in order to perform a fair comparison. Taking this issue into account and following the NIST guidelines for the selection of TLS implementations [90], only two pairs of TLS 1.2 cipher suites fulfill the requirements:
ECDHE-ECDSA-AES128-**CBC**-SHA256ECDHE-RSA-AES128-**CBC**-SHA256
and
ECDHE-ECDSA-AES128-**GCM**-SHA256ECDHE-RSA-AES128-**GCM**-SHA256

The main difference between these two options is the block cipher mode of operation: one uses Cipher Block Chaining (CBC), while the other one uses Galois/Counter Mode (GCM). Since the aim of the experiments is to provide future-proof conclusions, the latest TLS 1.3 draft [91] was evaluated. The draft indicates that only block ciphers with Authenticated Encryption mode of operation [92] will be allowed in TLS 1.3, being AES-GCM one of the chosen algorithms. Thus, the only cipher suites that accommodate all the mentioned restrictions are ECDHE-**ECDSA**-AES128-GCM-SHA256 and ECDHE-**RSA**-AES128-GCM-SHA256.

After selecting the cipher suites to be used and before generating the certificates, ECC curves and RSA key sizes have to be selected. The limiting factor for such a selection is the available ECC curves and the RSA key sizes supported by the ESP32 hardware and the ESP-IDF SDK. According to the official documentation, the hardware crypto-engine of the ESP32 SoC supports up to 512-bit ECC curves and up to 4096-bit RSA keys.

To take advantage of the hardware-acceleration capabilities of the ESP32, an ECC software implementation for each of the selected curves has to be available and ESP-IDF supports the ECC curves shown in Table 2. As explained in Section 2.6, NIST and SECG curves are much faster than the brainpool curves. Due to this fact, certificates were generated using all the NIST and SECG curves available on the ESP32. In addition, it was evaluated the brainpool BP256R1 for comparison purposes. Four RSA certificates with 1024-bit, 2048-bit, 3072-bit, and 4096-bit key sizes were generated. Please note that the 4096-bit RSA key will provide a security level of roughly 150 bits, although this is an approximate value since it is not one of the security levels considered traditionally by NIST (i.e., 80, 112, 128, 192, 256) and no equivalent value was found in the literature. Therefore, the 4096-bit RSA key size was considered in the tests as it is the largest key size theoretically supported by the ESP32 crypto-engine.

## 4. Experiments

### 4.1. Testbed Setup

To evaluate the performance of the selected devices, different tests were performed using both configurations of the testbed previously described in Section 3. The tests consisted of transferring a set of files from the mist end-device (ESP32 server) to the fog gateway (Orange Pi PC) and to the second mist end-device (ESP32 client) using an HTTPS connection. As it was explained in Section 3.2, the ESP32 server was programmed as an HTTPS server that allowed for accessing JSON files stored on the ESP32 flash memory. All the generated JSON files have a similar structure, varying in size from 32 bytes to 8 kilobytes. The JSON files were generated using a Python script and the text generation library Faker (version 0.7.3) [93]. The Python script allows for creating files with the desired payload size and with a structure similar to the payloads usually transmitted by IoT sensor nodes. To verify that the generated files were similar in terms of structure and data redundancy to the actual files transmitted in real IoT networks, they were compressed using GZIP and the compression obtained ratios were analyzed and compared to the values presented in [94].

Regarding the hardware-acceleration options available on the ESP32, both SHA and AES were enabled for all the tests, since both algorithms are used by the selected cipher suites. Multi-precision integer (MPI) acceleration was turned on only when using the RSA cipher suite. Although the official ESP32 documentation states that its crypto-engine supports RSA operations for a key size of up to 4096 bits, it was only possible to keep MPI acceleration enabled when using the 1024-bit and the 2048-bit RSA certificates for the HTTPS server implementation. When using the certificates generated with 3072-bit and 4096-bit RSA keys, it was only possible to establish an HTTPS communication with the MPI acceleration disabled. This is due to the fact that when the MPI acceleration was enabled, the ESP32 server rebooted itself as soon as the HTTPS communication was initiated. The exact reason of this issue remains unknown, being the most probable cause a software-related bug on the hardware-acceleration implementation. In contrast, the ESP32 client implementation allowed for maintaining MPI acceleration enabled for all the RSA key sizes.

The NIST modulo-*p* optimizations for ECC were enabled for both cipher suites. It is important to note that, as stated in [45], the ECC optimizations also impact the RSA cipher suite performance, since they accelerate the Elliptic curve Diffie–Hellman Ephemeral (ECDHE) key exchanges where ECC is also used.

A high-level sequence diagram of the testing procedure is depicted in Figure 5. Since the ARMbian distribution was installed without a graphical desktop environment, the laptop was used to establish an SSH connection to the Orange Pi PC to allow for the execution of a Python script. Such a script had a list with the different JSON payload sizes available on the ESP32. For each of the payloads, an individual test was executed. As it is shown in Figure 5, the first step consists in sending an HTTP request to the HTTP server that runs on the Orange Pi PC that is in charge of obtaining the energy measurements. When this request is received by the Orange Pi PC, it starts obtaining current samples from both INA219 sensors. At the same time, the fog gateway (or the mist end-device client, depending on the configuration of the testbed) starts to download the JSON file from the ESP32 server a total of 100 times. The time spent by each request is also registered. When all the downloads are finished, the client device sends another request to the energy meter (the second Orange Pi PC), which stops the measurement procedure and obtains the total energy consumption of the test. Finally, the energy consumption values are stored into a file along with a throughput value calculated considering the 100 file transactions and the total time spent in the test.

It is important to note that, although it can be considered that only two scenarios are evaluated (i.e., fog gateway to mist end-device and mist end-device to mist end-device communications) the use of different cipher suites and security levels gives as a result multiple combinations that can be seen as individual scenarios. For instance, for each of the testbed configurations, two cipher suites are used, and the selected RSA key sizes and ECC curves compare a total of 6 different security levels provided by 13 cipher suites configurations. Those security levels can be associated with different levels of data protection (e.g., non-confidential, confidential, restricted) and target different mist end devices, depending on their computational capabilities. Likewise, the obtained throughput values can be translated into different service needs, giving priority either to real-time actuation or to higher security assurance, depending on the application requirements.

### 4.2. Testbed Performance Analysis

As it was previously indicated, payloads from 32 bytes to 8 kilobytes were transmitted a total of 100 times for each of the tested RSA key sizes and ECC curves. As explained in Section 4.1, the testing procedure is initiated by the fog gateway by sending an HTTP request to the Orange Pi PC in charge of registering the energy consumption from the two INA219 sensors. Due to the nature of the involved HTTP communications, delays were expected during the start and stop of the measurement procedure. Therefore, the period during which the INA219 sensors values were recorded was slightly larger than the actual duration of the test. To measure the impact of this expected delay in the current measurement process, both Orange Pi PCs were synchronized using NTP [95]. Timestamps were then recorded at both Orange Pi PCs, which allowed for determining the mentioned start and stop time offsets. Table 3 presents the time difference average, taking into account all the performed tests for each ECC curve and RSA key size. The formula used for obtaining the presented values is:
(EnergyMeasurementTime−FilesDownloadTime)×100/EnergyMeasurementTime

It can be observed that the average time difference percentage considering all tests is 0.011%, which means that the current measurement period is only 0.011% larger than the actual test, so its impact is negligible in the experiments presented in this article.

### 4.3. Influence of Payload Size on Energy Consumption

To determine the impact of the payload transmitted when using the selected cipher suites, they were compared for the lowest security level of the ones evaluated in this article (80 bits). Such a security level was selected since it requires the lowest computational resources, thus allowing for determining in a clearer way than with higher security levels the impact on energy consumption and data throughput related exclusively to payload size.

Figure 6 shows the energy consumption values for each payload when using a 1024-bit RSA key and a secp192k1 ECC curve. As it can be observed, the size of the payload has little to no effect on the total energy consumption, only showing a slight increase on payloads over 4 kB. Almost identical figures were obtained for the rest of the tested RSA key sizes and ECC curves. These results were expected due to the computational capabilities of the evaluated devices.

It is important to note that most of the time and computational effort of the transmission are dedicated to the TLS Handshake, specifically to the signing and verification processes [29]. During such signing and verification processes the RSA and ECC keys of the generated certificates are used, thus resulting in a greater impact on the energy consumption than the actual transmission of the payload.

### 4.4. Comparative Analysis of the Provided Security Level, Energy Consumption and Data Throughput for the Selected Cipher Suites

The aim of this section is to present the results obtained by comparing the provided security level, the energy consumption and the data throughput of all the tested alternatives. As explained in the previous section, the payload effect on the energy consumption was constant through all the performed tests, being noticeable only for payloads larger than 4 kB. For this reason, the results presented in this section are only shown for a 512-byte payload, since the obtained results for different payloads are almost identical. Moreover, in the Figures shown in this section, the *X*-axis indicates the different security levels provided by each RSA key size and ECC curve as indicated in Table 2.

#### 4.4.1. Fog Gateway to Mist End-Device Scenario

Figure 7 shows the combined energy consumption of the mist end-device and the fog gateway for the different tested alternatives. The individual energy consumption of the ESP32 and the Orange Pi PC are represented by different shades and patterns. As it can be observed, RSA presents higher energy consumption compared to the ECC secp curves for all the security levels over 112 bits. The brainpool curve presents a larger energy consumption than the secp alternatives and even doubles the energy consumption of RSA for the same security level. Another interesting result is that r1 and k1 curves outperform each other depending on the security level. Moreover, when comparing the results for ECC curves for 112-bit and 128-bit security levels, it can be concluded that a curve providing a lower security level can present worse results, as observed in the case of the secp_r1 curves. This can be observed in Figure 8, where the throughput values are presented, where the secp256r1 curve provides better throughput than the less secure secp224r1 curve. The reason for this behavior is further discussed on Section 4.5.

#### 4.4.2. Mist End-Device to Mist End-Device Scenario

In this scenario two mist end devices communicate to each other, one acting as client and the other one as server. As it can be observed in Figure 9, although the absolute energy consumption is higher than in the fog gateway to mist end-device scenario, the differences between security levels and cipher suites remain. The throughput shows a similar trend too, as it can be observed in Figure 10.

The higher energy consumption in this scenario is due to the slower response of mist end devices in comparison to fog gateways. To measure the impact of these delays on the communications, the network sniffer tool Wireshark was used in two different tests for recording all the timestamped transmitted packets. The first test consisted on downloading the 512-byte JSON file from a Python server deployed on a virtual machine using the ESP32 client. On the second test, a Python client was used to download the same file from the ESP32 server. Table 4 presents the measured delays between the SYN and the SYN-ACK messages of the three-way handshake of the TLS protocol and the delay between the Server Hello Done and the Client Key Exchange messages of the TLS handshake depicted in Figure 11. As it can be seen, the Client Key Exchange message is sent by the client after receiving the Server Hello Done message and performing the needed cryptographic operations for generating the client session key, which makes the delay between these two messages a good approximation to determine how fast the client is.

As it can be seen in Table 4, when the Python client starts the communications, the delay of the ESP32 server is close to 300 ms, which is clearly higher than the 0.00003 s delay required by the Python server when responding to the ESP32 client. Moreover, it takes almost 1 s to the ESP32 client to respond with a Client Key Exchange message after receiving the Server Hello Done sent by the Python server, while the Python client responded on less than 2 ms to the ESP32 server.

In order evaluate the real-time performance of the presented testbed, the average time per request was measured on three scenarios: fog gateway to fog gateway, fog gateway to mist end-device, and mist end-device to mist end-device. As an example of evaluation of secure cipher suites, RSA 2048 and secp256r1 performance were compared. The obtained results are shown in Figure 12. As it can be observed, secp256r1 is faster than RSA 2048 in two of the three scenarios, while they spend the same amount of time (roughly 6 s) in the mist end-device to mist end-device scenario. Therefore, in this latter scenario it is not possible for the evaluated mist end-device to provide quick and secure responses.

### 4.5. Analysis of the Results

To determine the reliability of the obtained results, it is first necessary to analyze the reliability of the testbed. In this regard, the implemented software allowed for obtaining 500 samples per second from each sensor, which is a high enough sample rate to detect any energy consumption spikes or fast variations that occur during the tests. Moreover, the time synchronization between the two Orange Pi PCs of the testbed allowed for determining that the current measurement period was on average only 0.011% larger than the actual test. That means that for the longest test (of 581.41 s), the measurement procedure only required roughly 63 ms.

When analyzing the energy consumption and throughput results for the fog gateway to the mist end-device scenario, the BP256R1 curve presented the worst results of all the tested alternatives. As already discussed in Section 2.6, brainpool curve calculations are slower than the secp curves. In fact, for the same security level, the brainpool curve consumes twice the energy of the RSA-based alternative.

With respect to the secp curves, it can be concluded that both the random NIST curves and the Koblitz SECG curves behave similarly in terms of energy consumption for both the fog gateway and the mist end-device. They even provide almost the same throughput, outperforming slightly each other depending on the security level. An interesting finding is the reduction in energy consumption of the curve secp256r1 with respect to secp224r1, especially for the mist end-device: the former consumes less energy and provides better throughput than the later. This is due to the NIST *modulo p* optimizations enabled in the ESP32 (as explained in Section 4.1, these optimizations were enabled to accelerate ECC calculations). Such optimizations are curve and platform dependent, which implies that they must be programmed and refined for each hardware platform and curve.

It is also worth indicating that, when comparing RSA with ECC, it can be observed that, as the security level increases, the required key sizes for ECC grow in a more linear way than the ones required for RSA. ECC also presents less energy consumption and better throughput for each of the tested security levels. For instance, for a 128-bit security level, RSA consumes twice the energy of both the secp256r1 and the secp256k1 curves. The performance difference is even more noticeable as the required security level increases since, for a security level of 256-bit, secp521r1 consumes less energy than 3072-bit RSA, which in comparison, only provides 128 bits of security.

Finally, when comparing the mentioned results to the ones obtained in the mist end-device to mist end-device scenario, it can be concluded that they behave almost in the same way in terms of relative performance between ECC and RSA. However, in this second scenario, the total energy consumption is three times larger and the throughput five times lower in some cases. This degradation in performance is explained due to the slower response time of the ESP32 modules in comparison to the Orange Pi PCs, so the delays between the different TCP and TLS messages are increased.

Finally, it is worth noting the slow response time observed in Section 4.4.2 for the mist end-device to mist end-device scenario, which is due to the large delays introduced in the different phases of the communications protocols (i.e., during the TCP and TLS handshakes). As a consequence, real-time services cannot be provided with the evaluated mist end-device hardware, so more powerful devices and software optimizations on the HTTPS server and clients would have to be performed. One of such optimization would consist in the use of long-living connections such as websockets, which are able to reduce the number of TCP and TLS handshakes and to allow for faster communications on the mist layer, thus providing real-time or near real-time capabilities.

## 5. Conclusions

In this paper, it was analyzed the difference on energy consumption and data throughput of different RSA and ECC-based cipher suites for securing the HTTP communications between a mist end-device and a fog gateway and between two mist end devices. A real-world testbed was implemented and evaluated, allowing for obtaining up to 1000 current samples per second. The deviation between the tests duration and the energy measurement period of the testbed was measured, and with only a 0.011% deviation, it was considered to have no effect on the presented results. Two different hardware platforms were selected with different hardware and software capabilities in order to be used as mist end devices and as fog gateways. Two different cipher suites (i.e., ECDHE-ECDSA-AES128-CBC-SHA256 and ECDHE-RSA-AES128-CBC) were chosen following the NIST guidelines and following the restrictions of the latest TLS draft. For each cipher suite different alternatives for providing a range of security levels were analyzed and selected. Certificates were generated and tested using different ECC curves and RSA key sizes.

After analyzing the results, it can be concluded that, in the performed tests, ECC is a better alternative than RSA for mist deployments, since it presents less energy consumption and better throughput. The energy consumption values obtained for the mist end devices and the fog gateway are always lower for the ECC certificates than for the RSA ones when compared at the same security level. The only exception for the previously conclusion was the brainpool ECC curve, which presented the worst results of all the tested alternatives. When comparing the secp ECC curves, they achieve similar performance at the same security level. An important finding regarding ECC curves is the platform and curve dependency on the implemented optimizations. In this scenario, the most secure secp256r1 curve presented better results than the secp224r1 curve. This fact reinforces the idea of the need for real-world testing when throughput and energy consumption are critical for a deployment. Thanks to the performed experiments, it was demonstrated empirically that relying on the size of the curve alone could end up on selecting less secure curves with lower throughput and higher energy consumption values.

Regarding the evaluation of the communications between two mist end devices, the performed tests showed higher energy consumption and lower throughput values than in the case of the fog gateway to mist end-device communication scenario. Specifically, the reason behind this degradation in performance is communications delay, which derived into up to three times larger energy consumption values and up to five times lower throughput results. Therefore, the results of the tests performed on the mist layer indicate that further work must be done by future mist computing developers in order to improve the mist layer communications performance.

## Figures and Tables

**Figure 1 sensors-18-03868-f001:**
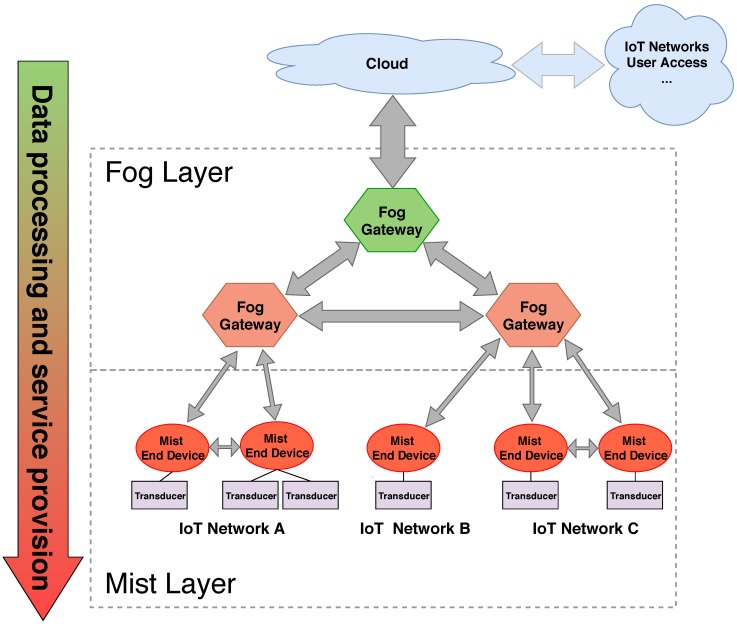
Generic IoT edge computing architecture with fog and mist computing layers.

**Figure 2 sensors-18-03868-f002:**
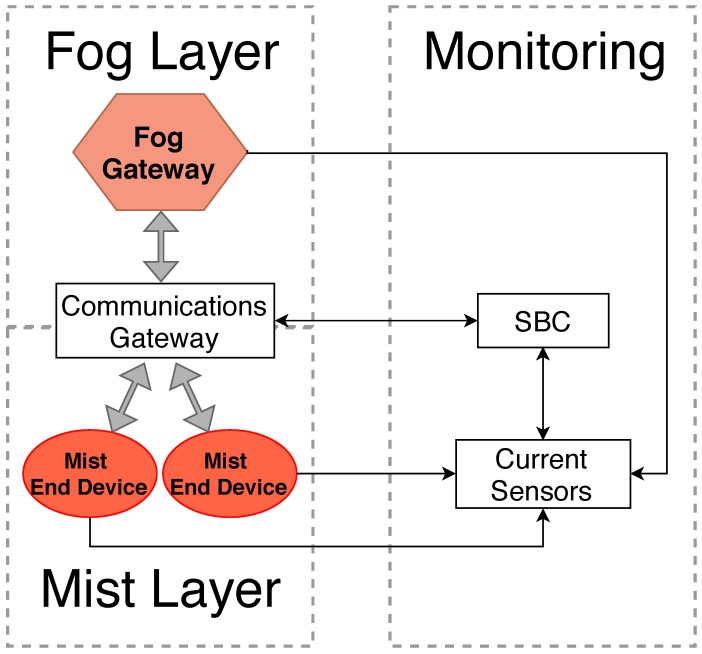
Testbed architecture.

**Figure 3 sensors-18-03868-f003:**
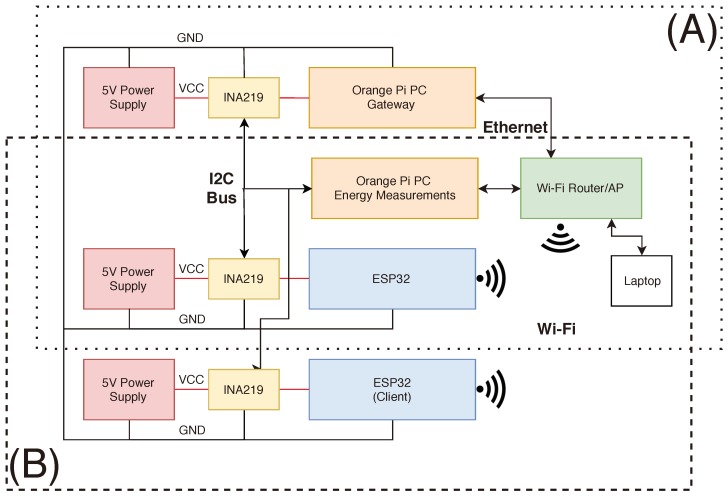
Testbed architecture configurations. Fog Gateway to Mist End-Device (**A**) and Mist End-Device to Mist End-Device (**B**).

**Figure 4 sensors-18-03868-f004:**
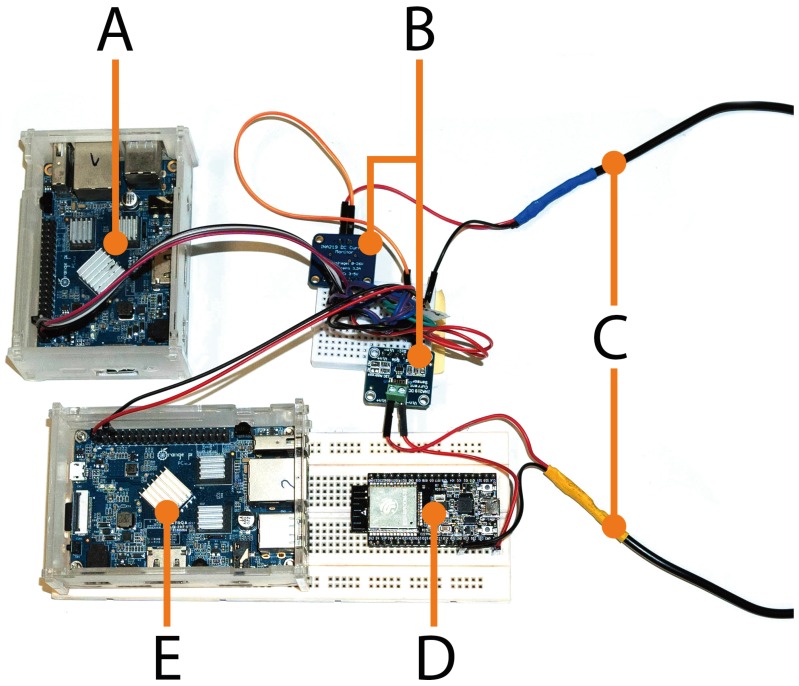
Main components of the testbed. (**A**) Orange Pi PC for performing energy measurements; (**B**) INA219 current sensors; (**C**) 5 V power supply; (**D**) ESP32 (mist end-device); (**E**) Orange Pi PC (fog gateway).

**Figure 5 sensors-18-03868-f005:**
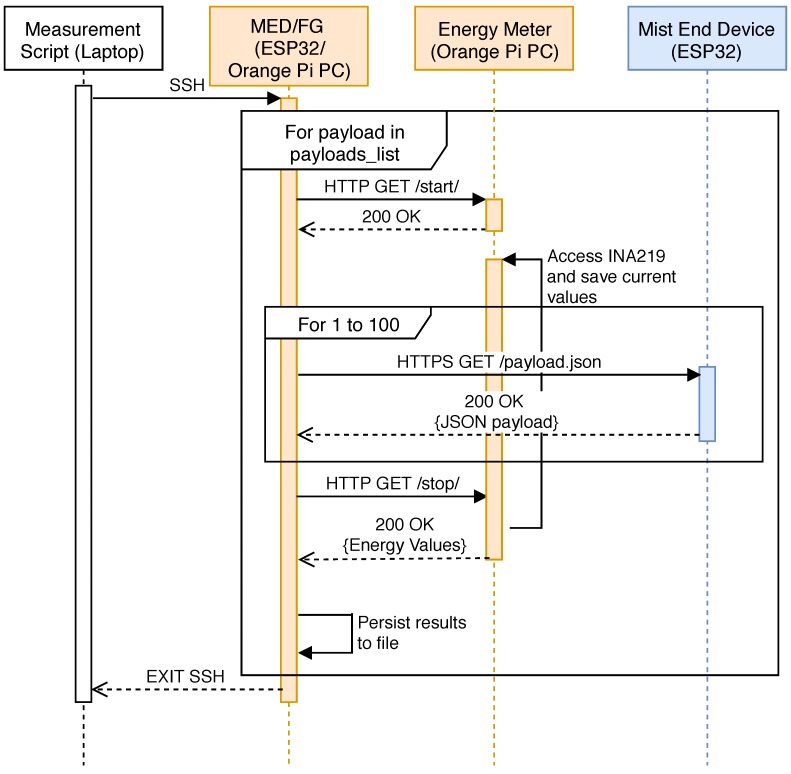
Sequence diagram of the testing procedure.

**Figure 6 sensors-18-03868-f006:**
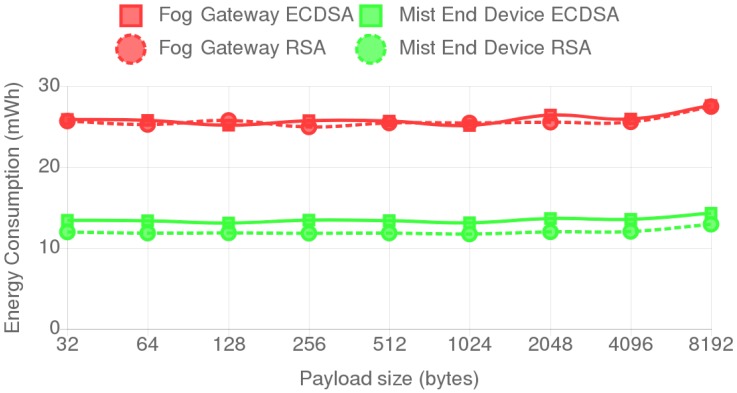
Energy consumption for all the tested payloads when using a 1024-bit RSA key and a secp192k1 curve.

**Figure 7 sensors-18-03868-f007:**
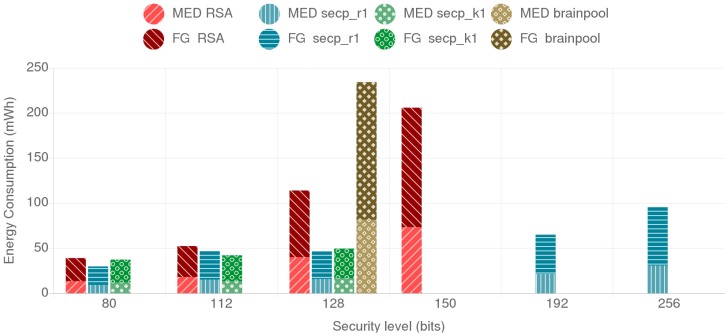
Fog Gateway (FG) and Mist End-Device (MED) combined energy consumption.

**Figure 8 sensors-18-03868-f008:**
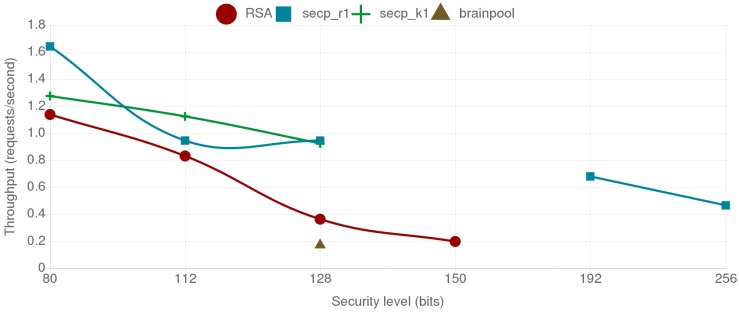
Throughput values in requests per second between the mist end-device and the fog gateway.

**Figure 9 sensors-18-03868-f009:**
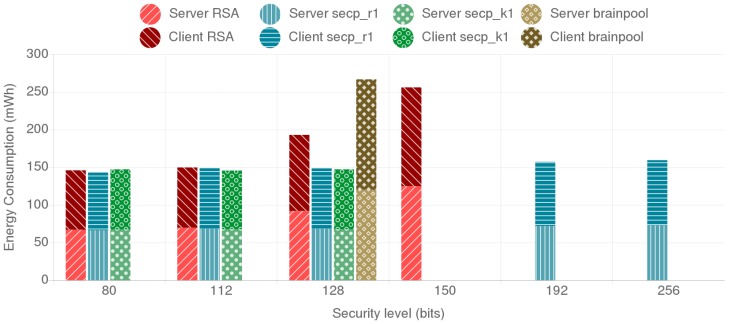
Mist end-device server and mist end-device client combined energy consumption for the different tested alternatives.

**Figure 10 sensors-18-03868-f010:**
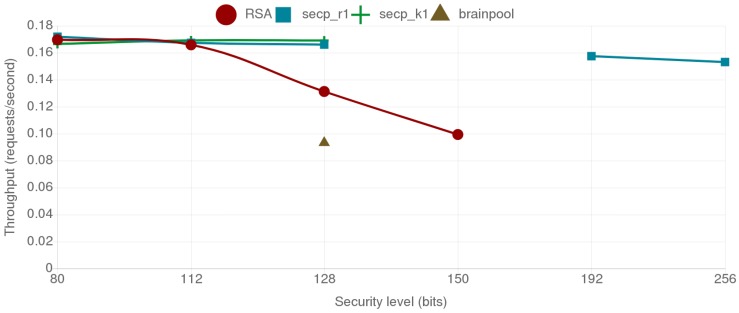
Throughput values in requests per second between both mist end devices.

**Figure 11 sensors-18-03868-f011:**
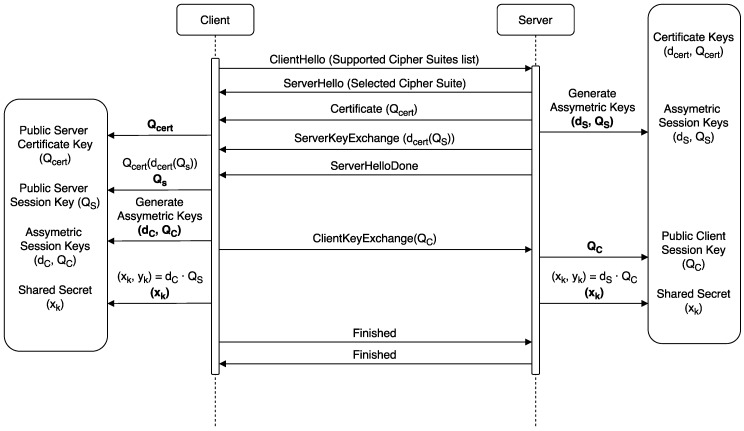
TLS Handshake procedure for ECDHE-ECDSA-AES256-GCM-SHA384 and similar cipher suites [29].

**Figure 12 sensors-18-03868-f012:**
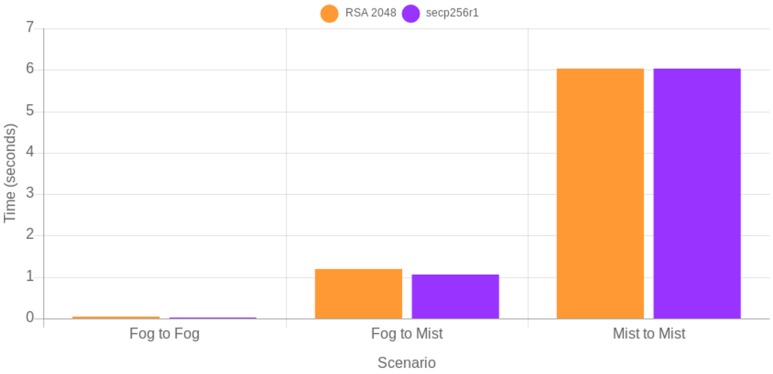
Time (seconds) per request for fog gateway to fog gateway, fog gateway to mist end-device, and mist end-device to mist end-device scenarios.

**Table 1 sensors-18-03868-t001:** Main characteristics of popular IoT development boards that can be used for implementing mist end devices.

Name	Clock Rate	Cores	RAM Size	Max. Power	Connectivity	References
Intel Edison Module *	500 MHz	2	1 GB	1 W	Wi-Fi/BLE	[36]
Arduino Tian	560 MHz/48 MHz	1/1	64 MB/32 KB	1.551 W	Wi-Fi	[37]
Arduino MKR WiFi 1010	48 MHz	1	32 KB	2.31 W	Wi-Fi/BLE	[38]
LightBlue Bean+	8 MHz	1	2 KB MB	3.5 W	BLE	[39]
ESP32	240 MHz	2	512 KB	1.65 W	Wi-Fi/BLE	[40]
Particle Photon	120 MHz	1	128 KB	1.419 W	Wi-Fi	[41]

* Discontinued.

**Table 2 sensors-18-03868-t002:** Comparable security levels for RSA and ECDSA.

Security Level	RSA Key Size	ECDSA Key Size	ESP-IDF Curves
80	1024 bits	160–223 bits	secp192r1, secp192k1
112	2048 bits	224–255 bits	secp224r1, secp224k1
128	3072 bits	256–383 bits	secp256r1, secp256k1, bp256r1
192	7680 bits	384–511 bits	secp384r1, bp384r1
256	15,360 bits	512+ bits	secp521r1, bp521r1

**Table 3 sensors-18-03868-t003:** Time difference (in percentage) between the energy measurement process and the file download process for the fog gateway to mist end-device scenario.

RSA Key Size or ECC Curve	Time Difference (%)
rsa1024	0.016
rsa2048	0.013
rsa3072	0.005
rsa4096	0.003
secp192k1	0.018
secp192r1	0.023
secp224k1	0.016
secp224r1	0.013
secp256k1	0.013
secp256r1	0.015
bp256r1	0.003
secp384r1	0.009
secp521r1	0.006

**Table 4 sensors-18-03868-t004:** Relevant TLS delays in mist end-device to mist end-device communications.

	ESP32 Server	ESP32 Client
SYN to SYN-ACK	**0.27745**	0.00003
Server Hello Done to Client Key Exchange	0.00189	**0.97986**

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
