# Peer review of "A Practical Evaluation on RSA and ECC-Based Cipher Suites for IoT High-Security Energy-Efficient Fog and Mist Computing Devices"

_sensors, 2018, doi:10.3390/s18113868_

Round 1

Reviewer 1 Report

The authors present a theoretical study proposing some architectural components to deal with security issues within the Internet of Things era. Specifically, the authors introduce a high-security energy-efficient fog and mist computing architecture and a corresponding testbed. The testbed makes use of Transport Layer Security (TLS) Elliptic Curve Cryptography (ECC) and Rivest-Shamir-Adleman (RSA) cipher suites (that comply with the yet to come TLS 1.3 standard requirements), which are evaluated and compared in terms of energy consumption and data throughput for a fog gateway and two mist end devices. The obtained results allow the authors to conclude that ECC outperforms RSA in both energy consumption and data throughput for all of the tested security levels. The topic of the paper is interesting and well-aligned with the scope of the journal. Furthermore, the authors have very well structured their paper, which enables the reader to easily follow the proposed analysis, and they have well-thought-out their main contributions. However, the authors should address the following comments in order to improve the quality of their paper and its presentation:

1.       One of the main questions that arises from the whole manuscript is how they security threats in the Internet of Things networks are detected by the authors? Based on which characteristic? Do the authors consider characteristics, such as their transmission behavior, transmission power, etc. There is a great part of the recent literature, where the security threats are detected based on transmission behavior, e.g., Tsiropoulou, Eirini Eleni, John S. Baras, Symeon Papavassiliou, and Gang Qu. "On the Mitigation of Interference Imposed by Intruders in Passive RFID Networks." In International Conference on Decision and Game Theory for Security, pp. 62-80. Springer, Cham, 2016, Sagduyu, Yalin Evren, Randall A. Berryt, and Anthony Ephremidesi. "Wireless jamming attacks under dynamic traffic uncertainty." In Modeling and Optimization in Mobile, Ad Hoc and Wireless Networks (WiOpt), 2010 Proceedings of the 8th International Symposium on, pp. 303-312. IEEE, 2010, Sagduyu, Y.E. and Ephremides, A., 2009. A game-theoretic analysis of denial of service attacks in wireless random access. Wireless Networks, 15(5), pp.651-666, etc. The authors have missed to review this methodology, which is fundamental in the field of the security within the IoT era. The authors should update section 2 and the references list accordingly.

2.       In section 3, the authors should clarify if the proposed testbed architecture can be applied in any type of heterogeneous network within the IoT era, such as dense networks consisting of femto-access points, pico-access points, etc. instead of the wifi router. Is the extension of this work to other types of communication environments in the IoT networks straightforward? Should adaptations and other assumptions be considered? The authors should clarify this point and provide the holistic aspect of their proposed analysis.

3.       In section 4, the authors provide comparative results practically only with one comparative scenario. The authors should provide additional comparative scenarios, as currently the provided comparison is very narrow.

4.       Within the whole manuscript, the authors avoid to discuss the complexity analysis, the implementation cost and time complexity of the proposed framework. How feasible is this framework to be implemented in a real-time or close to real-time? The latter is crucial characteristic for the security threats in the Internet of Things networks and the authors should provide some indicative numerical results and corresponding discussion.

Overall, this is a very interesting paper, with novel ideas and the authors should carefully address the provided comments to improve their manuscript.

Author Response

Dear Sir/Madam,

Please find attached our detailed responses to the comments. 

Regards.

Reviewer 2 Report

The paper presents an evaluation regarding RSA and ECC asymmetric ciphers for IoT high-security energy-efficient fog and mist devices. The paper is very interesting and well written. There are two minor issues I see: 

a) the use of "an" instead of "a" in four places (lines 270, 281, 367 and 389); and 

b) there are two many keywords (maybe the second and the fifth can be deleted).

Author Response

(The authors gave the same response as above.)
